# Global Positioning System (GPS) Scintillation Associated with a Polar Cap Patch

**Jayachandran P. Thayyil** [1,2,*], **Anthony M. McCaffrey** [1], **Yong Wang** [2], **David R. Themens** [3], **Christopher Watson** [1], **Benjamin Reid** [1], **Qinghe Zhang** [2] and **Zanyang Xing** [2]

1. Physics Department, University of New Brunswick, Fredericton, NB E3B 5A3, Canada; a.mccaffrey@unb.ca (A.M.M.); chris.watson@unb.ca (C.W.); Ben.Reid@unb.ca (B.R.)
2. Shandong Provincial Key Laboratory of Optical Astronomy and Solar-Terrestrial Environment, Institute of Space Sciences, Shandong University, Weihai 264209, China; wangyong180@sdu.edu.cn (Y.W.); zhangqinghe@sdu.edu.cn (Q.Z.); xingzanyang@sdu.edu.cn (Z.X.)
3. Space Environment and Radio Engineering Group (SERENE), School of Engineering, University of Birmingham, Birmingham B15 2TT, UK; david.themens@unb.ca
* Correspondence: jaya@unb.ca; Tel.: +1-506-447-3330

**Abstract:** A Global Positioning System (GPS) network in the polar cap, along with ionosonde and SuperDARN radar measurements, are used to study GPS signal amplitude and phase scintillation associated with a polar cap patch. The patch was formed due to a north-to-south transition of the interplanetary magnetic field (IMF Bz). The patch moved antisunward with an average speed of ~600 m/s and lasted for ~2 h. Significant scintillation occurred on the leading edge of the patch, with smaller bursts of scintillation inside and on the trailing edge. As the patch moved, it maintained the integrity of the scintillation, producing irregularities (Fresnel scale) on the leading edge. There were no convection shears or changes in the direction of convection during scintillation events. Observations suggest that scintillation-producing Fresnel scale structures are generated through the non-linear evolution of the gradient drift instability mechanism.

**Keywords:** polar cap ionosphere; plasma irregularities; ionospheric scintillation; solar-terrestrial interaction; GNSS technology

## 1. Introduction

Ionospheric scintillation, the stochastic variations observed in the amplitude and phase of a trans-ionospheric radio signal, is generated by Fresnel scale irregularities in the ionosphere [1,2]. Historically, the scintillation phenomenon was first observed in signals from radio stars [3] and was later studied in detail using satellite radio beacons [2]. Scintillation research has gained prominence recently because of the need for mitigating/forecasting the effect of scintillation on Global Navigation Satellite System (GNSS)-based applications, effects of which include the reduction in accuracy or complete loss of positioning capacity [4]. Forecasting a stochastic phenomenon such as scintillation is a daunting task, and the only way one can achieve even limited success is through an understanding of the underlying mechanism(s) and the conditions under which the scintillation is produced. Recent GPS scintillation studies, based on receiver-generated scintillation indices, have concentrated on the occurrence and morphology of scintillations in the polar region [5–8]. One notable finding of these studies has been the dominant occurrence of phase scintillation (as opposed to amplitude scintillation) in the polar region. However, recent studies [9,10] questioned the validity of this result. In the polar region, the ionospheric drift is much higher compared to lower latitudes, which makes the traditional choice of a 0.1 Hz fixed cutoff frequency a poor choice in detrending the GNSS phase measurements. This choice allows for refractive variations in phase to contaminate the results (due to the inclusion of structures in the phase that are lower than the Fresnel frequency). A recent study [11]

revealed that there exist fluctuations that are refractive in nature that can mimic phase scintillations, while [10] showed that dominant phase variations in the polar and auroral regions are indeed refractive. Recently, studies [12,13] also emphasized the importance of proper detrending in the identification of scintillation in the polar region.

More than 60 years of scintillation research has revealed several morphological features [14,15] and led to the development of many possible theoretical formalisms [1,16,17] to describe scintillation producing irregularities. However, the underlying mechanism(s) that produces scintillation, especially in the polar cap region, is still not known. Dynamical behavior of the polar cap ionosphere is primarily driven by magnetospheric convection and neutral circulation and undergoes structuring over a wide range of temporal and spatial scale sizes. This structuring is due to the interplay of mechanical forces, electrodynamics, and ionization chemistry [18–28]. Spatial scales of irregularities vary from thousands of kilometers to a few centimeters and the scale sizes that produce scintillation (Fresnel scale) are of the order of a few hundred meters at F-region altitudes. Previous studies [25,29–34] and references therein have shown the coexistence of scintillation-producing structures within polar cap patches and auroral forms. Theoretical and numerical simulation studies have shown that these small-scale structures are produced on the trailing edge of convecting polar cap patches through the gradient drift instability and shear instability mechanisms [18,19,35] and penetrate to the leading edge [36]. However, these theories and simulations are still not well tested, and recent observations [25,30] have provided contradictory evidence, as no preferred geometry of the ray paths concerning electric field and gradients for the scintillation occurrence and scintillation occurred on both the leading and trailing edges of patches.

Amplitude scintillation occurrence in the polar region is very low due to the nature of the scintillation producing irregularities [37]. It is compounded by completed dominance of phase variations due to the problems with the detrending of the phase of the GNSS signal [12,13] to determine the phase scintillation index. However, the introduction of the new dynamic detrending and method along with the calculation of the modified scintillation method [38] provided an opportunity to detect amplitude scintillation occurrence accurately. In this paper, we present a case study of GPS signal amplitude and phase scintillation observation associated with a polar cap patch in an attempt to identify the underlying mechanism(s) that generates radio wave scintillation in the polar region.

## 2. Data, Methods of Analysis, and Observations

We used GPS Canadian Advanced Digital Ionosondes (CADI) of the Canadian High Arctic Ionospheric Network (CHAIN) [39]. For the details of the CHAIN instruments and locations, please see [36]. We also used SuperDARN convection measurements [40] to get contextual ionospheric drift velocity during the event studied. The patch event we considered for this study occurred on 7 March 2010 between 04:00–07:00 UT. Figure 1 shows the location of the CHAIN ionosonde and GPS receivers along with the location of the Ionospheric Pierce Point (IPP) of GPS satellite (PRN) ray paths at seven CHAIN stations, mapped using the ionosphere altitude of 270 km (the approximate ionospheric peak height measured by the ionosondes). The figure also indicates the time of initial increase in the slant GPS Total Electron Content (TEC) associated with the patch detected by respective GPS ray paths (color-coded) at seven CHAIN stations, as shown in Figure 1 below. The letter after each PRN denotes the station identification (E for Eureka, R for Resolute Bay, P for Pond Inlet, T for Taloyoak, H for Hall Beach, C for Cambridge Bay, and I for Iqaluit). A TEC increase associated with the patch was first detected at ~04:18 UT by PRN 24 at Eureka and last by PRN 7 at Iqaluit at ~05:31 UT. The figure clearly illustrates that the TEC increase associated with the patch moved more or less antisunward.

We could infer that the patch was formed due to a sudden southward turning of the interplanetary magnetic field (IMF) at ~03:58 UT (Figure 2). IMF Bz was predominantly northward until the sudden transition at 03:58 UT. Even though times are corrected for the travel time from the upstream solar wind monitor to the magnetopause, the correction

has a certain level of uncertainty [41]. To avoid this uncertainty, a detectable ionospheric response (e.g., changes in the direction of convection) was taken as the time of arrival of the IMF changes in the ionosphere [42].

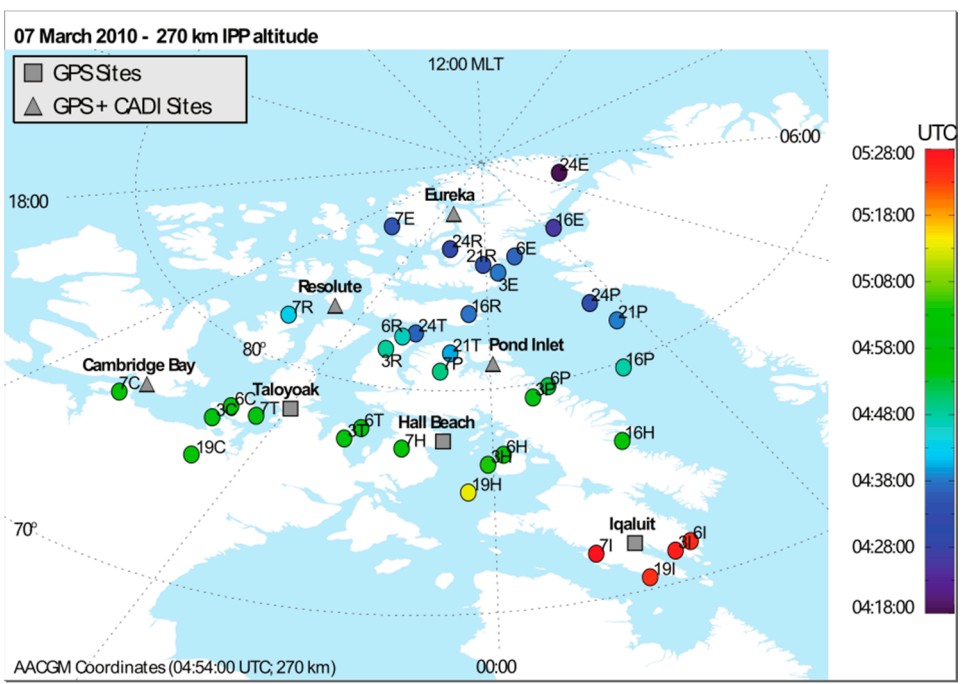

**Figure 1.** Location of CHAIN GPS and CADIs along with the Ionospheric Pierce Point (IPP) location of GPS ray paths of several polar cap stations, at time of initial patch detection on 7 March 2010. A height of 270 km, based on four ionosonde measurements, is used for the calculation of the IPP location. The letter after each PRN represents the station name (E for Eureka, R for Resolute Bay, P for Pond Inlet, T for Taloyoak, C for Cambridge Bay, H for Hall Beach, and I for Iqaluit). The time of the arrival of the signature of the patch at each IPP location is color-coded. For example, the TEC increase associated with the patch is first observed by PRN 24 at Eureka at 04:18 UT.

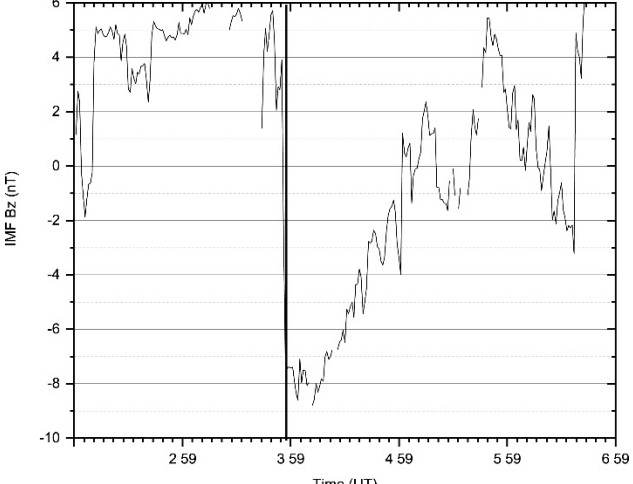

**Figure 2.** IMF Bz for the interval 02:00–07:00 UT of 7 March 2010 from the OMNI database. Time was corrected for the travel time from the solar wind monitor to the magnetopause. This correction is an estimate. The IMF southward transition occurred around 03:58 UT (marked by a vertical line). The IMF was predominantly northward before the transition.

Convection measured by CADI at Eureka and Resolute Bay (Figure 3) shows that the convection direction changed from sunward to antisunward and convection speed started to increase rapidly around 04:10 UT at both stations (marked by vertical lines in the figure). This type of convection response to the north-south transition of the IMF was previously detected and reported [43]. Therefore, the time of arrival of the IMF change at the ionosphere is ~04:10 UT. The patch arrived at Eureka around 04:32 UT and Resolute around 04:37 UT. This delay relative to the change in convection is expected since the patch is transported from the dayside cusp through the polar cap at the convection speed, while the convection response to the IMF Bz transition was almost immediately after the IMF transition.

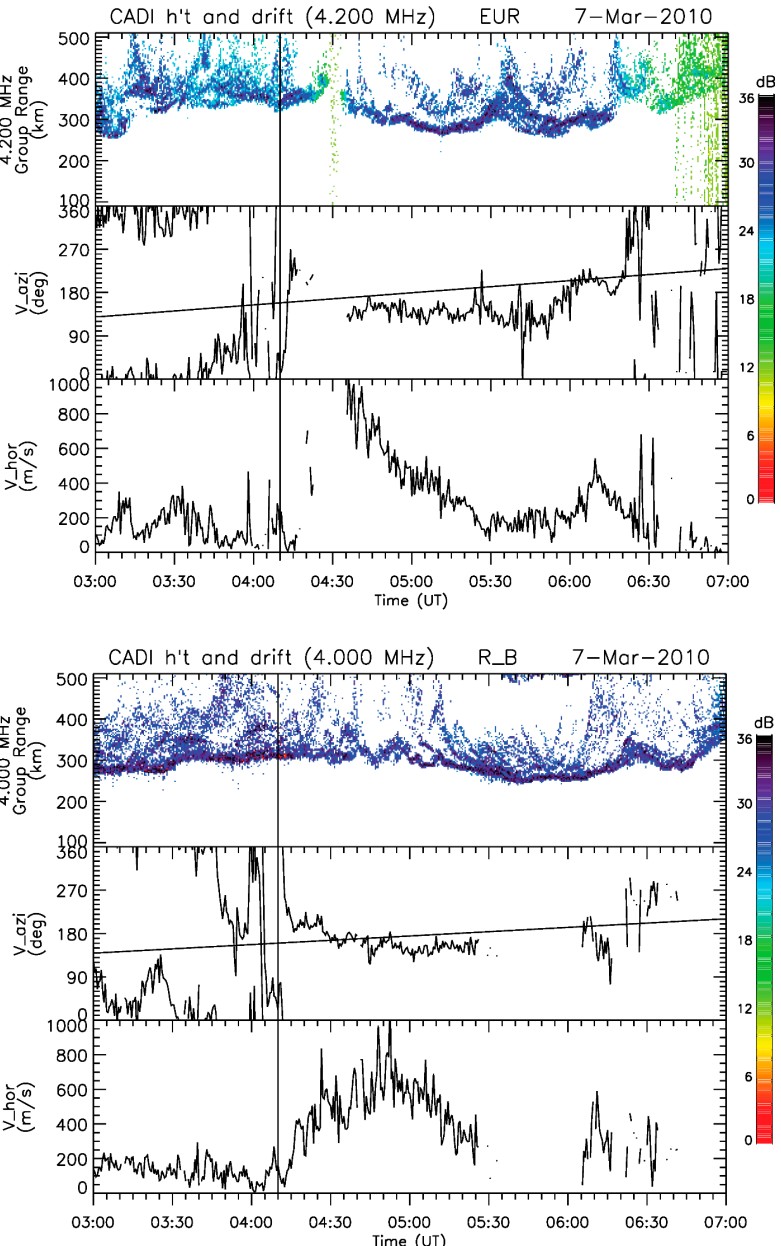

**Figure 3.** CADI measurements from Eureka (**top panel**) and Resolute (**bottom panel**) for the time interval 03:00–07:00 UT of 7 March 2010. Each panel shows, from top to bottom, group range, convection direction, and convection speed. The expected antisunward convection direction for each station is marked by a solid line in each convection direction plot.

Figure 4 shows the change in TEC (ΔTEC) at seven CHAIN GPS stations (left panel) [44] along with the critical frequency of the F-layer (foF2) at 4 ionosonde stations of CHAIN (right panel) during the interval 04:00–07:00 UT. ΔTEC is in TEC units (1 TECU = $10^{16}$ ele/m$^2$). It can be seen from the foF2 variations that the patch signature is first observed at Eureka at ~04:32 UT, followed by Resolute Bay at ~04:37 UT Pond Inlet at ~04:48 UT, and Cambridge Bay at ~04:53 UT. This is consistent with the patch motion that can be interpreted from Figure 1. If we take 04:10 UT (convection change) as the time of arrival of the IMF north-south transition at the ionosphere, it took nearly 22 min for the patch to form and travel at the convection speed from the patch generation (source) region (probably near the cusp located approximately at 75° MLT around 11:00 MLT) to the center of the polar cap (Eureka). The time delays between the four-ionosonde stations are also consistent with this scenario.

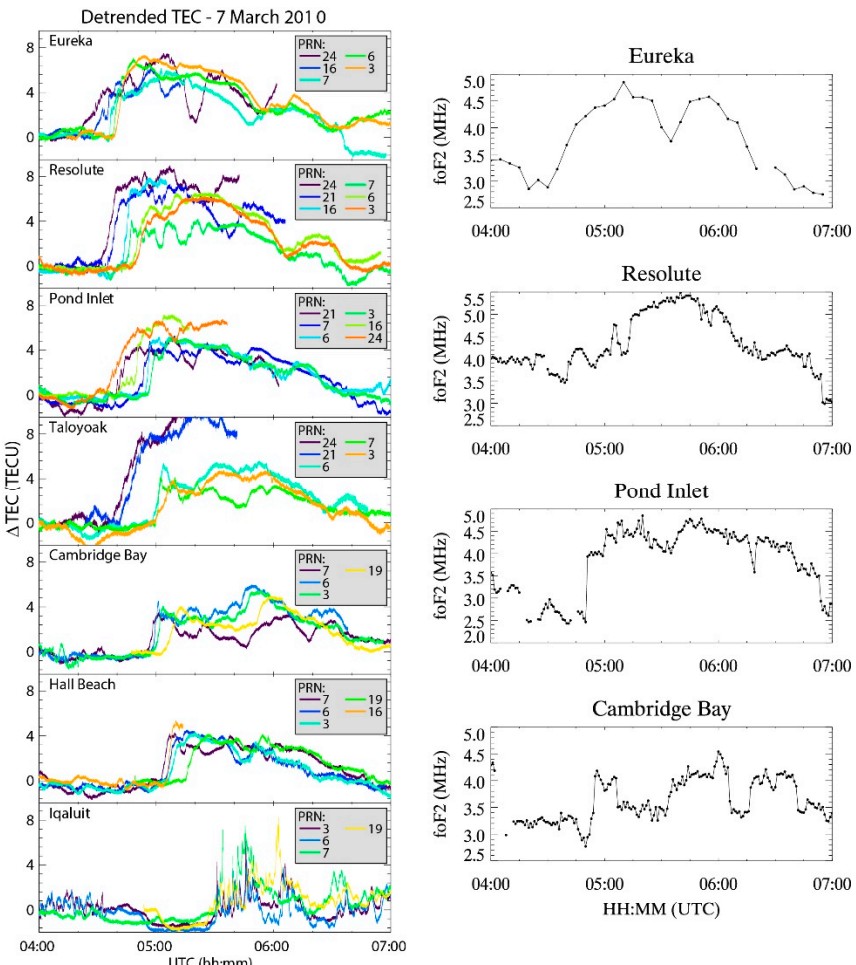

**Figure 4.** The critical frequency of the F layer measured by four ionosondes (**right panel**) and changes in TEC measured by seven GPS receivers (**left panel**) for the time interval 4:00 –07:00 UT on 7 March 2010. This figure along with Figure 1 clearly shows the patch signature and its motion in the antisunward direction.

The first signature of the patch in the TEC (an increase in TEC), was detected along the ray path of PRN 24 at Eureka around 04:18 UT (this is earlier than the ionosonde time because the IPP location of PRN 24 is closer to the source region), and progressively appeared on other ray paths and at other stations as the patch moved antisunward. The time of initial increase in TEC associated with the patch corresponds to the time indicated in Figure 1 for each corresponding IPP. Please keep in mind that these are slant TEC values, and the variations of TEC along ray paths are different because of the geometry of the ray path with respect to the patch. For example, the increase in TEC is more gradual along PRN

24 and 16 at Eureka, and more sudden along other ray paths. The average amplitude of the TEC increase associated with the patch also decreased as the patch moved antisunward (~7.5 TECU at Eureka and Resolute Bay to 4 TECU at Hall Beach and Cambridge Bay). The time delays between TEC signatures clearly show the motion of the patch is antisunward, consistent with the foF2 variations. The patch lasted for around 2 h. The speed of the patch, calculated using the GPS TEC triangulation technique [45], is ~600 m/s. The average convection speed during the patch at Eureka and Resolute Bay (measured by CADIs) is around 650 m/s (Figure 2), and the average convection speed measured by SuperDARN radars was about 680 m/s (please see the discussion section).

The TEC and foF2 measurements clearly show that there was a patch and that the patch moved antisunward with the convection. The patch had gradients of differing magnitude in the direction of motion, as well as perpendicular to the direction of motion (from the rate of change of TEC along different ray paths), which can generate Fresnel scale irregularities that can cause scintillation. Figure 5 shows the variation of detrended amplitude (left panel) and phase (right panel) along the PRN 7 ray paths measured at the seven CHAIN GPS stations. Figure 5 is arranged in descending latitudes from top to bottom (Eureka at the top and Iqaluit at the bottom). We used 50 Hz data for this analysis, which was detrended using the method described in the literature [38]. The raw detrended time series of amplitude and phase variations has a high probability of multi-path contamination, which can mimic scintillation, especially at high latitudes where the elevation angle can be lower. In order to avoid this, we used the day before and the day after the event to compute modified scintillation indices [38] for the patch event in this study. For the details of the calculation of modified scintillation indices, please see the original paper [38]. The modified scintillation indices are plotted as red solid lines in Figure 3 under raw detrended amplitude and phase time series. By taking this approach, we removed almost all of the multi-path contamination in the data. For example, a close examination of the amplitude time series of Eureka shows several candidate events for scintillation after 06:00 UT. However, the corresponding modified scintillation index does not show any variation at all after 06:00 UT, suggesting that those events are due to multi-path effects. Orange and green dashed lines in each figure represent fluctuation in the modified indices of one and two standard deviations from the mean, respectively. We chose any fluctuation above one standard deviation as scintillation. We also chose to use PRN 7 measurements as an example because it was one of three PRNs that were recorded at all stations continuously for the entire duration of the event. Note that the first signature of the patch along the PRN 7 ray path at Eureka was observed at ~4:35 UT and that the initial foF2 signature associated with the patch at Eureka was around 04:32 UT. It can be seen from the figure that bursts of amplitude and phase scintillation associated with the patch started around 04:35 UT along the PRN 7 ray path at Eureka, with scintillation observed at progressively later times at lower latitudes. A closer look at Figures 4 and 5 reveals that the onset of scintillation along the PRN 7 ray path at each station started around the same time as the initial TEC enhancement associated with the patch detected along the same ray path. Both amplitude and phase scintillation bursts associated with the patch are more or less observed at all the stations, with a systematic delay that is consistent with the TEC enhancements. Another interesting feature evident from Figures 4–6 is that the Fresnel scale irregularities in the patch, which produced the scintillation, were mainly confined to the leading edge of the patch, with smaller bursts inside and on the trailing edge. It is also interesting to note that scintillation producing irregularities in the leading edge of the patch maintained their integrity as the patch convected across the polar cap.

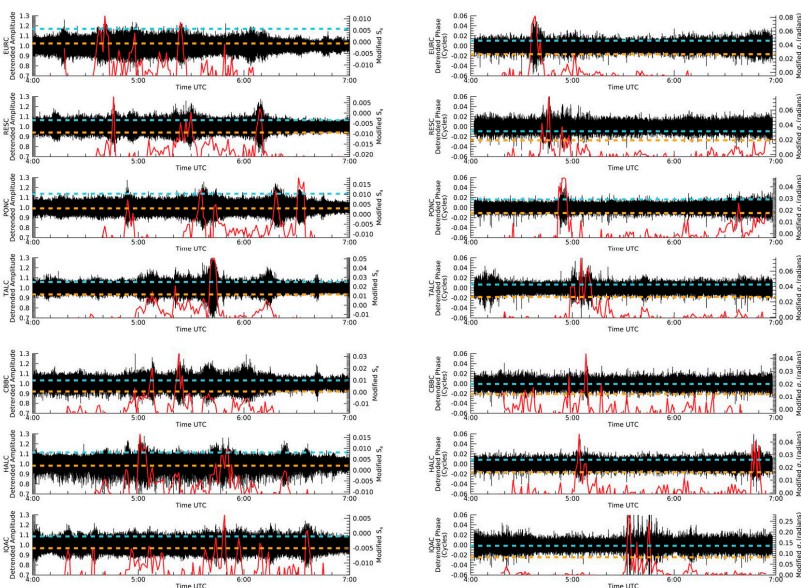

**Figure 5.** Raw detrended amplitude (**left panel**) and phase (**right panel**) variations of the GPS PRN 7 signal measured at seven GPS stations. The figure is arranged in descending latitudes from top to bottom (Eureka at the top row and Iqaluit at the bottom). Both amplitude and phase scintillation can be seen at all the stations, starting first at Eureka around 04:35 UT (the same time as the initial patch TEC change) and seems to propagate in the same direction as the patch. The red solid line in each figure shows the modified scintillation index, while orange and green dashed lines represent 1 and 2 standard deviation levels, respectively, from the mean.

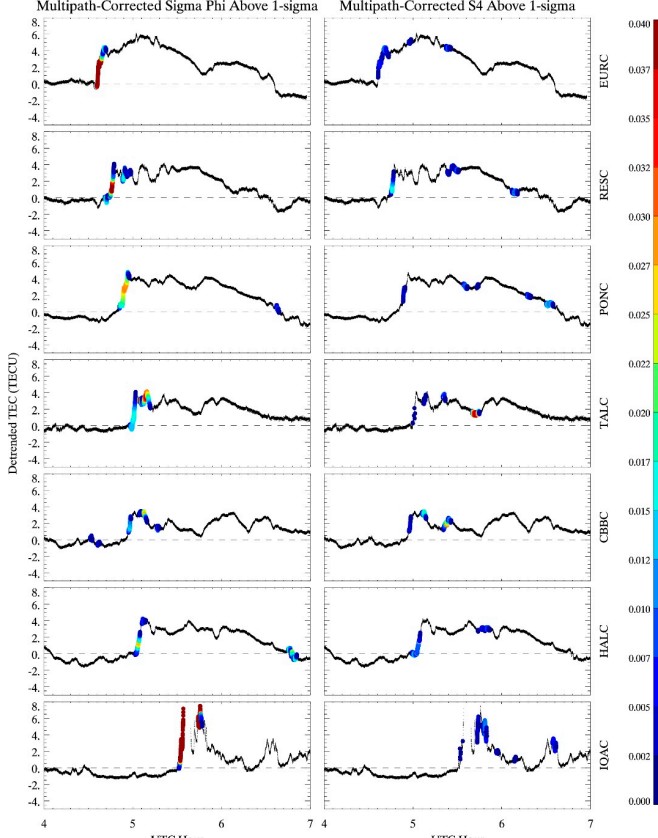

**Figure 6.** ΔTEC variations (black lines) and modified amplitude and phase scintillation indices (colored circles) from PRN 7 at all seven GPS stations.

To make this point clearer, Figure 6 shows the time series of PRN 7 ΔTEC (black lines) and modified scintillation indices above the 1-sigma threshold (colored circles) for all seven GPS stations. This figure clearly shows that most of the scintillation-producing irregularities are confined to the leading edge of the patch and that they more or less maintained their integrity as the patch convected across the polar cap. Note that the PRN 7 measurements at Iqaluit had some cycle slip issues.

To more extensively illustrate the occurrence of scintillation producing irregularities associated with the patch for the period 4:00–07:00 UT, Figure 7 shows the magnetic latitude (AACGM) and local time of amplitude (top panel) and phase (bottom panel) scintillation detected by all PRNs for all seven GPS receivers. IPP trajectories at 270 km altitude are shown as solid lines (color-coded by PRN), while square symbols indicate the presence of scintillation along each trajectory. Scintillation was considered to occur when the modified scintillation index exceeded one standard deviation above the mean. The purpose of this figure is twofold: (1). To show that amplitude and phase scintillation were present on ray paths other than those of PRN 7; (2). Scintillations were present on ray paths of different geometry with respect to the patch motion (along and perpendicular to the motion). This confirms our prior observation of the presence of amplitude and phase scintillation on the leading edge of the patch, with no preferred GPS signal geometry/direction for the scintillation to occur.

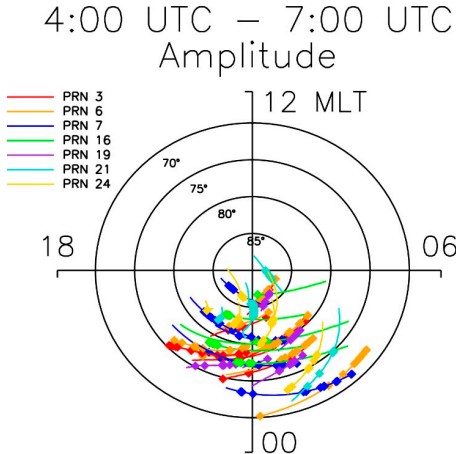

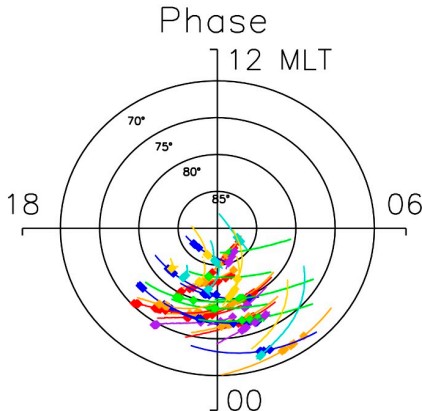

**Figure 7.** Magnetic coordinates of IPP trajectories (solid colored lines) for all PRNs observed by all seven GPS receivers, along with occurrence of amplitude (**top**) and phase (**bottom**) scintillation (colored squares). Color coding is according to PRN.

## 3. Discussion

Three important observations based on this study are (1). Both amplitude and phase scintillation producing irregularities associated with a polar cap patch were present; (2). Significant scintillation was confined to the leading edge of the patch, with smaller bursts inside and on the trailing edge of the patch. (3). As the patch moved across the polar cap, the integrity of the scintillation producing irregularities (Fresnel scale) in the leading edge was more or less maintained, indicating that Fresnel scale irregularities are not generated locally. This study confirms the earlier observation of scintillation on the leading edge of the patch [30].

Scintillation, which is a diffractive phenomenon, requires the presence of Fresnel scale irregularities in the ionosphere [1,16,17]. The two main plasma instability mechanisms that can generate these scintillation-producing structures are the gradient drift instability [18] and Kelvin–Helmholtz/shear instability [46]. There are certain preferred geometries and/or conditions for these irregularities to occur. For example, in the F-region, the classical linear-gradient drift instability mechanism requires that density gradients be parallel to the direction of the plasma drift (or electric field perpendicular to the gradient), while the shear instability requires the presence of strong flow shears. For the linear gradient drift instability, the requirement for the density gradient and flow direction to be aligned ensures that growth rates are greatest at the trailing edge of density structures; thus, scintillation would primarily occur in the trailing edge of the patch. Since our observations show the confinement of scintillation primarily to the leading edge of the patch, the linear gradient drift instability mechanism is not responsible for the production of Fresnel scale irregularities in this case. Figure 8 shows representative convection measured by the SuperDARN radars at six representative time intervals during the event. The seven red stars denote the locations of CHAIN GPS stations. The area covered by the CHAIN GPS receivers is under a streamlined high-speed flow, with no strong shears or changes in the observed convection. The convection was roughly antisunward and shows more or less a laminar flow (with varying flow speed with time) in the region of interest. The lack of shear flows or changes in the direction of convection in the region of observed scintillation also suggests that a mechanism other than the classical shear instability mechanism was responsible for irregularity generation. Please keep in mind that the range resolution of the radar is 45 km and there may be possibilities of shear flows that are smaller in size than the resolution of the radar. There is a possibility of flow shears moving with the patch, which can create flow shear instability [47]. Scintillation-producing Fresnel scale irregularities were present on the leading edge of the patch when it first appeared on the GPS signal ray path at Eureka and those structures remained intact as the patch moved across the polar cap. Maintenance of the integrity of Fresnel scale irregularities on the leading edge of the patch suggests that the irregularities are not produced by local shears or local conditions at these individual observing locations. In addition, through a close examination of Figures 1 and 4–6, one can see that amplitude and phase scintillations were present throughout the leading edge of the patch, without any preferred GPS signal path geometry with respect to the convective flow and patch motion. This again suggests that these irregularities are not produced by local conditions as the patch is convected across the polar cap. Therefore, none of the local linear theories mentioned above can fully explain our observations. A possible explanation of the observations presented is the 3-D nonlinear simulation work published [36]. Their simulation study showed the presence of structures with scale sizes of 100 m–10 km (which includes Fresnel scales that are responsible for producing scintillation) on both the leading and trailing edges, as well as inside the patch. They have attributed this structuring to either the nonlinear development of the gradient drift instability or to the occasional reversal of the direction of the convection. However, in the case studied in this paper, the convection remained more or less in the antisunward direction. Therefore, in this case, the structuring was most likely due to the nonlinear evolution of the gradient drift instability mechanism. The work [35] also mentioned that the most important feature of the nonlinear 3-D evolution is that the patch remains intact even later after the

patch is fully structured. Figures 4 and 5 confirm this aspect well. Even though most of the observed features of the patch structuring can be reasonably well explained using the numerical simulation study [35], one observation that does not fit is the time scale in which these Fresnel scale structures are produced. According to the simulation, full structuring of the patch takes more than an hour, whereas observations presented in this paper show evidence of full structuring within 10–20 min after the patch was generated by the north-south transition of the IMF. This discrepancy needs to be investigated further to fully understand the structuring of the patches.

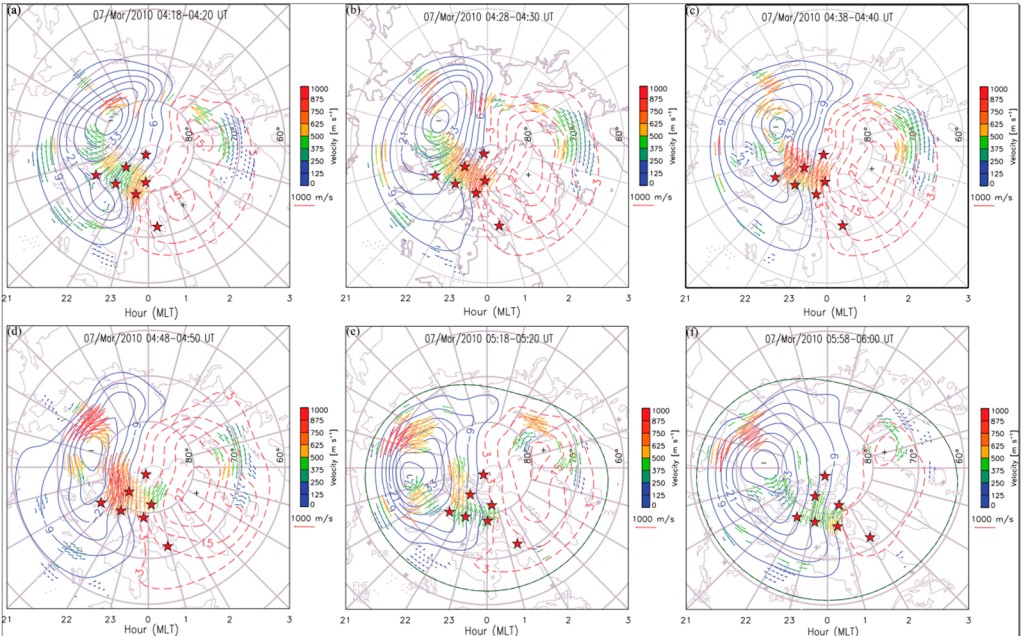

**Figure 8.** Representative high-latitude convection measured by SuperDARN radars at four-time intervals during the event. Locations of the seven CHAIN GPS stations are marked on the figures as red star symbols.

## 4. Conclusions

A case study of polar cap patch dynamics and associated GPS signal amplitude and phase scintillation, using multi-instruments, yielded the following:

1.  Most of the amplitude and phase scintillation occurrences were confined to the leading edge of the patch with a limited occurrence inside and trailing edge of the patch.
2.  The integrity of the scintillation producing irregularities (Fresnel scale) was maintained as the patch convected across the polar cap.
3.  Scintillation occurrence does not depend on the geometry of the GPS ray path concerning the patch.
4.  Although the above results seem to indicate that the scintillation is produced through the non-linear evolution of the gradient drift instability mechanism, a thorough modeling approach that can resolve the Fresnel scale is required to fully understand the generation mechanism of radio wave scintillation in the polar region.

**Author Contributions:** J.P.T. conceptualized the study and wrote the manuscript. A.M.M. completed the scintillation analysis and created Figures 5 and 7. D.R.T. completed the ionosonde data analysis and created Figure 6. C.W. completed the TEC analysis and created Figures 1 and 3. Y.W. completed the SuperDARN data analysis and created Figure 8. B.R., Q.Z. and Z.X. contributed to the scientific discussion and all the authors read and commented on the manuscript. All authors have read and agreed to the published version of the manuscript.

**Funding:** This research received no external funding.

**Institutional Review Board Statement:** No animals or human beings are used in this study. Therefore, this is not applicable.

**Informed Consent Statement:** Not applicable.

**Data Availability Statement:** CHAIN data is available through http://chain.physics.unb.ca/chain/ (accessed on 30 March 2021). SuperDARN data is available through http://vt.superdarn.org (accessed on 30 March 2021). Solar wind and interplanetary magnetic field data are available through https://omniweb.gsfc.nasa.gov/ (accessed on 30 March 2021).

**Acknowledgments:** Infrastructure funding for CHAIN was provided by the Canadian Foundation for Innovation and the New Brunswick Innovation Foundation. CHAIN operations are conducted in collaboration with the Canadian Space Agency. This research was undertaken with the financial support of the Canadian Space Agency FAST program and the Natural Sciences and Engineering Research Council of Canada. The authors also wish to thank the International Space Science Institute in Beijing (ISSI-BJ) for supporting and hosting the meetings of the International Team on "Multiple instrument observations and simulations of the dynamical processes associated with polar cap patches/aurora and their associated scintillations", during which the discussions leading/contributing to this publication were initiated/held.

**Conflicts of Interest:** The authors declare no conflict of interest. The funders had no role in the design of the study; in the collection, analyses, or interpretation of data; in the writing of the manuscript, or in the decision to publish the results.

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
