# Peer review of "Global Positioning System (GPS) Scintillation Associated with a Polar Cap Patch"

_remotesensing, doi:10.3390/rs13101915_

Round 1

Reviewer 2 Report

Using digisonde and GPS receiver, this paper reported GPS scintillations inside a polar patch, which can be harmful to normal GPS communications. The author then concluded that scintillation-producing Fresnel scale structures are generated through the non-linear evolution of the gradient drift instability. Here I think the paper needs major revision before it is suitable for publication.

Major comments:

1 In the whole introduction part, although the author pointed out the disadvantages of those previous studies, they did not emphasize their own advantages here in this study. The author shall provide several sentences in the end of introduction part to point out their advantages compared with previous studies

2 Some of the figures are not well set.

2.1 Figure 2  I think the author can provide extended range of Bz variations, maybe expand to March 6 to March 8. Furthermore, the author shall also give other geomagnetic indicies to show how the geomagnetic activity is during that focused period

2.2 Figure 3 description is too redundant. From ‘the convection direction’ in Line 121 to line 126 had better to be put into paper main text, not just in the figure description

2.3 Figure 5 is too crowded. Maybe consider just put one locations results in the figure and put other locations into Appendix?? And the black line is no need to be presented since the red line is the modified scintillation index.

3 The discussion and conclusion had better be separated. For the conclusion part, please provide the conclusion from the paper one by one, not just put them together

4 Since there is a network, why not put these into a real map, which may be clearer and easier to read and follow the structures? Have you checked the TEC map from Madrigal database to compare??

Minor comments:

Line 14  I think the author means the southward turning of IMF Bz, is that right?  If it is, please revise IMF into IMF Bz

Line 28 replace [12] with ]1-2]

Line 44 replace ‘Recenet’ with ‘Recent’

Line 128-129, this sentence is redundant, please remove , ‘determined using the technique developed by our research group [41] (left panel)’. There is no need to mention how this is derived

Line 209 replace ‘more clear’ with ‘clearer’

Round 2

Reviewer 2 Report

The revision improved the paper and solve all my previous concerns. I recommend this paper to be published in present form